# Combined Use of Cointegration Analysis and Robust Outlier Statistics to Improve Damage Detection in Real-World Structures

**DOI:** 10.3390/s22062177

**Published:** 2022-03-10

**Authors:** Simone Turrisi, Emanuele Zappa, Alfredo Cigada

**Affiliations:** Politecnico di Milano, Department of Mechanical Engineering, Via La Masa 1, 20156 Milano, Italy; emanuele.zappa@polimi.it (E.Z.); alfredo.cigada@polimi.it (A.C.)

**Keywords:** structural health monitoring, novelty detection, cointegration, robust outlier analysis, environmental and operational variations, sensor data fusion, natural frequencies, acceleration signals

## Abstract

Due to the need for controlling many ageing and complex structures, structural health monitoring (SHM) has become increasingly common over the past few decades. However, one of the main limitations for the implementation of continuous monitoring systems in real-world structures is the effect that benign influences, such as environmental and operational variations (EOVs), have on damage sensitive features. These fluctuations may mask malign changes caused by structural damages, resulting in false structural condition assessment. When damage identification is implemented as novelty detection due to the lack of known damage states, outliers may be part of the data set as the result of the benign and malign factors mentioned above. Thanks to the developments in the field of robust outlier detection, the current paper presents a new data fusion method based on the use of cointegration and minimum covariance determinant estimator (MCD), which allows us to visualize and to classify outliers in SHM data, depending on their origin. To validate the effectiveness of this technique, the recent case study of the KW51 bridge has been considered, whose natural frequencies are subjected to variations due to both EOVs and a real structural change.

## 1. Introduction

The main engineering disciplines associated with damage detection are undoubtedly structural health monitoring (SHM) and non-destructive evaluation (NDE). Even though the most recent NDE techniques have proven to be very effective in case of composite laminates or pipe inspection [1], SHM procedures offer many advantages when applied to the case of real-world civil structures. First, SHM is based on permanent sensors, while in NDE, the devices (i.e., eddy-current probe, X-ray generator, etc.) are usually brought to the desired point of the structure. Therefore, SHM can be conducted in real-time, or at least very frequently, in an automated fashion, while NDE is performed sporadically or on request. Second, the choice of SHM philosophy allows us to install a limited amount of inexpensive sensors to obtain reliable measurements; on the other hand, the local nature of the physical effects exploited in NDE requires us to install a dense layout of transducers to reach an adequate coverage area.

Dealing with data-based approaches to SHM, damage sensitive features are extracted from these measurements and machine learning methods are used to build a statistical data model [2]. In general, damage detection procedures exhibit better.

Performances within supervised learning scenarios, where class labels (i.e., damage levels and locations) are identified in advance. Unfortunately, for the majority of practical SHM applications it is not possible to obtain known damage state data, but only data from conditions assumed as “normal”. In this case, the SHM implementation relies on unsupervised learning methods, such as novelty detection, which only require data from the undamaged state (baseline) of the structure [3]. Then, a statistical model of the normal conditions is obtained based on the feature distributions and new data are tested for conformity to the model.

The idea of novelty detection using outlier analysis is not a new one and many SHM application examples can be found [4,5,6]. This well-established statistical method essentially seeks for the presence of outliers in the data, i.e., observations that appear inconsistent with the rest of the data set and therefore are believed to be generated by an alternate mechanism. A classic discordancy measure used in the previous studies to reveal the presence of outliers is the Mahalanobis squared distance (MSD) [7]. MSD indicates how far a generic point is from the cloud points center of mass, taking into account the shape of the cloud as well. However, the major limitation of such a distance is that it suffers from outliers “masking effect”. Given that MSD is characterized by sample mean and covariance, any outliers present in the baseline data set have a negative influence on the estimation of mean and covariance. This increases the occurrence of false negatives, since new observations associated with small distances could instead hide the presence of outliers. Within a SHM context, a potential issue is represented by the influence that environmental and operational variations (EOVs) have a structure features. This effect is particularly evident for civil infrastructures such as bridges, whose measured responses are constantly subjected to important variations induced by wind, humidity, traffic, human loading [8] and, above all, temperature [9,10,11,12]. Since changes due to these benign reasons typically manifest with the same order of magnitude of those due to structure degradations, they could appear as outliers in the data masking any potential indication of ongoing structural damage.

To solve the problem of inclusive outliers in the training data, robust statistical methods are considered in a wide range of researches, including SHM [13,14,15], to provide unbiased estimates of mean and covariance parameters computed from a smaller subset of data whose behaviour is assumed to be close to the true population values. Alternatively, data normalization techniques such as principal component analysis [9], autoencoders [16] or cointegration [17,18,19,20,21], are adopted to project the data into a different space to remove or at least reduce the effect of environmental and operational changes. Although previous methods proved their effectiveness in creating a normal condition training set clear from the influence of external factors, they are unable to distinguish which of the outliers indicated in the data are “benign” and which are “malign”. This distinction would represent a significant advancement for SHM practices, as it would make possible to identify external benign variations at an early stage, remove them from the analysis, so leaving the algorithm free to detect only real structural anomalies. Within SHM, the hierarchy of the damage identification process can be classified into four levels [22]: damage detection (level 1), damage localization (level 2), damage quantification (level 3) and damage prognosis (level 4). Different techniques are usually applied according to the desired SHM level. Recent advances in the SHM field offer methods, such as the Mode Shape Curvature or the Curvature Damage Factor [23], that can achieve up to level 4; however for civil structures it is usually accepted that the procedure is not optimized for such level of detail, due to the high complexity of the system which requires a trade-off between the level of the diagnostic system and the expense of training it adequately. Level 1 is also distinguished from the others, as it can be accomplished without the knowledge of the structure behaviour when it is damaged. This is why the goal of this paper is to propose a novel approach based on the use of cointegration and robust outlier statistics aiming at improving the damage detection strategies for real-world structures. Furthermore, as the techniques applied at the different levels mentioned above are found to be complementary, the possibility of having an efficient algorithm characterized by accurate detection and low computational complexity allows us to improve also the higher levels information of the SHM process.

Later in this paper it will be shown how, thanks to the mutual use of cointegration and robust outlier analysis, the underlying nature of outliers is revealed, allowing one to separate the effects due to environmental and operational factors out of those due to damage. Into details, cointegration is first used to create a linear combination of the original measured variables $u=\beta_{1}y_{1}+\beta_{2}y_{2}+\ldots +\beta_{n}y_{n}$, where u is referred to as cointegration residual, in which all the effects of EOVs are deleted. Other recent successful implementations of cointegration to eliminate environmental and operational trends from damage sensitive features can be found in [24,25,26,27,28]. The expression of the cointegration residual can equivalently be reformulated as a linear relationship where a generic variable $y_{i}$ is selected as response and the remaining variables becomes the regressors $x_{i}$, i.e., $y_{i}=b_{1}x_{1}+b_{2}x_{2}+\ldots +b_{n}x_{n}+u_{i}$, thus enabling to predict the behaviour of $y_{i}$. Since to distinguish between outliers and leverage points one has to consider the response $y_{i}$ in parallel with the regressors $x_{i}$, the minimum covariance determinant (MCD) estimator is then adopted to compute the robust discordancy measures of the regressors. They indicate the distance of a certain observation belonging to the space of input features from a region associated with the bulk of the data. The combination of the two previous tools lead to a new diagnostic representation in which the cointegration residual is plotted against the robust distances calculated by applying the MCD estimator on the input variables of the model. This kind of visualization permits to automatically classify the observations according to the process which originates them.

To the best of authors’ knowledge, the issue of characterizing and separating the effects of multiple outliers within the context of SHM has only been presented by Dervilis et al. in [29]. This work illustrates the use of robust regression analysis and robust discordancy measures to demonstrate that environmental and operational conditions can manifest themselves differently, when compared to damaged conditions. With respect to this study, the use of cointegration used in this work in place of a simple linear regression offers important advantages, especially in the case of real-world structures. Here, the number of features increases and the external factors affecting these features are often not measurable or known a priori. Even in the case in which temperature is found to be the dominant factor, the existing relationships between temperature and structural response are hard to establish, since they require us to manage complex phenomena such as non-uniform solar irradiations, thermal inertia, changes in the material properties and boundary conditions. Therefore, if all the influencing factors are not properly included in the model, linear regression may produce unreliable predictions in the response. In addition, SHM data often exhibits a non-stationary behaviour, mainly induced by EOVs, which may cause the issue of spurious regressions. Cointegration represents a viable solution to overcome these limitations as it is a technique which does not strictly require the direct measurement of EOVs, but only a set of non-stationary time series representing the output of a given process.

The presence of actual damage for in-service structures is something that is hard to find. That is why damage detection strategies are often validated using a numerical model of the structure, where different damage scenarios are simulated to reproduce realistic damage conditions [30,31,32]. Alternatively, many works rely on the use of experimental data from the well-known Z24 bridge benchmark study [33,34,35], where long-term continuous monitoring took place during the year before demolition and progressive failure tests were performed. In order to increase the availability of validation cases related to different types of real structures and damages, the method proposed in this paper considers the experimental data from the monitoring campaign of the KW51 railway bridge in Leuven [36] over a period of 15 months, recently acquired by the research group from the Structural Mechanics Section of the KU Leuven University.

This data set represents a challenging situation, as it includes data subjected to the contemporary presence of both changing environmental conditions and different structural states, i.e., before, during and after an operation of retrofitting to resolve a construction error that emerged during inspections. Since structural modifications associated with the retrofit manifest, among others, with changes in natural frequencies having a similar variability of those due to environmental parameters [37], the effect of confounding influences is here rather critical. Furthermore, a very limited number of publications inherent to this case study are currently available. In [38], the authors developed an automatic procedure to identify natural frequencies and strain mode shapes of the bridge deck starting from the strain time histories measured by a dense network of fiber-optic Bragg Grating sensors. Using these modal data, the influence of temperature fluctuations and that of retrofitting are extensively investigated. In [39], indeed, natural frequencies are used to evaluate the performance of linear regression and robust principal component analysis as damage detection techniques. The ability to remove variations due to environmental changes as well as to identify actual changes in the structural behaviour introduced by retrofitting are discussed.

The layout of the present paper is organized as follows. First, an introduction to the theoretical background on robust outlier detection and cointegration are explained in Section 2 and Section 3, respectively. Then, Section 4 describes the proposed approach for outliers discrimination, based on the combined use of cointegration and minimum covariance determinant. After this, Section 5 presents the main characteristics of the KW51 bridge, which represents the case study analyzed in this work. Finally, in Section 6, the technique is tested on the experimental data from the KW51 bridge and the expected performances are evaluated on this real database.

## 2. Robust Outlier Detection

### 2.1. Minimum Covariance Determinant (MCD)

Outliers are observations that do not follow the pattern of the majority of the data. For multivariate point clouds, outliers become difficult to detect because we can no longer rely on a visual perception of the problem. A classical method to quantify the presence of outliers is to compute the Mahalanobis Squared Distance (MSD), which is given by the following equation:(1)$${\mathrm{MSD}}_{i}^{2}={\left(z_{i}-\mu_{z}\right)}^{T}\;\Sigma^{-1}\left(z_{i}-\mu_{z}\right)$$
where $z_{i}$ is the generic multivariate observation, $\mu_{z}$ designates the mean of the sample observations and Σ is the sample covariance matrix. The MSD indicates how far $z_{i}$ is from the center of the distribution of the training data (location), taking into account the shape of the distribution as well (scatter).

The main disadvantage of the traditional MSD as expressed in Equation (1) is that it is sensitive to the “masking effect” of multiple outliers. Since the MSD is characterized by sample mean and covariance, any outlier present in the training data set will have a relevant influence on the estimation of mean and covariance; in this way, small distances could be assigned to new observations, resulting in increased chances for false negatives. This problem is frequent in many SHM applications, where the lack of a priori knowledge for the damaged conditions does not allow to exclude the outliers from the analysis at a preliminary stage. Consequently, the use of more robust methods for the estimation of mean and covariance in presence of inclusive outliers is of significant interest.

The pioneering work in this area was proposed in [40], in which a robust estimation of the covariance matrix was provided by iteratively removing subsets of data with the highest MSDs, until the covariance converged to within a desired tolerance. That estimator, however, has a breakdown value which is strictly related to the data dimensionality. The breakdown value represents the smallest fraction of observations that can significantly bias the estimated parameter from its true population value. Some years later, Rousseeuw [41] proposed the minimum volume ellipsoid (MVE) method, which attempts to find an ellipse of minimum volume that encloses a subset of data, under the assumption of normally distributed multivariate features. If MVE is found, then outliers can be detected as points outside the boundaries of the ellipsoid. Although it has a 50% breakdown value, it is computationally not so efficient and, therefore, was replaced by a superior estimator called minimum covariance determinant (MCD) [42], which has the same breakdown value, but provides the possibility to implement more efficient algorithms for its computation [43,44]. In the present paper, the FAST-MCD algorithm is adopted, whose implementation details can be found in [45]. The code used in the analysis for the computation of the MCD estimator was provided by a MATLAB library of robust statistical methods called LIBRA [46]. Readers can find an explanation of the main steps of the FAST-MCD technique in the following.

Suppose that a multivariate dataset is stored in an n×p matrix $\left[Z\right]\;=\;{\left(z_{1},\ldots ,z_{n}\right)}^{T}$ where $z_{i}={\left(z_{i1},\ldots ,z_{ip}\right)}^{T}$ is the ith observation, so n stands for the number of observations and p for the number of variables. Robust estimates for the location *μ*_MCD_ and the scatter $\left[\Sigma_{\mathrm{MCD}}\right]$ of $\left[Z\right]$ can be obtained by the computation of the MCD. The goal of the MCD method is to select n/2<h≤n observations whose classical covariance matrix has the lowest possible determinant, i.e., the lowest possible variance of the data. The raw MCD location estimate is then the average of these h points, whereas the raw MCD estimate of scatter is their covariance matrix, multiplied by a consistency factor. Since the MCD estimates can resist up to $\left(n-h\right)$ outliers, the breakdown value h, or equivalently the ratio α=h/n, determine the robustness of the estimator. When a large number of outliers is expected, h should be chosen close to αn with α=0.5. Otherwise, an intermediate value for h, such as 0.75n, is recommended to obtain a higher finite-sample efficiency. When p/n is small, some data points can become co-planar, making the outlier detection problem hardly feasible. Therefore, it is recommended that when n/p>5 to choose α=0.5. Finally, the MCD estimates of location and scatter is affine equivariant, meaning that they stay the same under affine transformations in the data. This makes them immune to different variables scales and rotations.

The exact MCD estimator is hard to compute, as it requires an exhaustive investigation of all h-subsets out of n. The authors of [45] developed the FAST-MCD algorithm, based on a Concentration step (C-step) which avoids such a complete enumeration. The main computation steps are described here. Given $\left[Z\right]\;=\;{\left(z_{1},\ldots ,z_{n}\right)}^{T}$, let $H_{1}\;\subset \;\left(1,\ldots ,p\right)$ be a subset of size h. Then, *μ*_1_ and $\Sigma_{1}$ represent the mean and the covariance matrix of the data in $H_{1}$. If $\left|H_{1}\right|\;\neq \;0$, the relative distances are calculated as:(2)$$d_{1}{\left(i\right)}^{2}={\left(z_{i}-\mu_{1}\right)}^{T}\;\Sigma_{1}^{-1}\left(z_{i}-\mu_{1}\right)$$

Then, $H_{2}$ is chosen such that $\left\{d_{i}\left(i\right);\;i\;\in \;H_{2}\right\}\;=\;\left\{{\left(d_{1}\right)}_{1:n},\ldots ,{\left(d_{1}\right)}_{h:n}\right\}$ where ${\left(d_{1}\right)}_{1:n}\;\leq \;{\left(d_{1}\right)}_{2:n}\;\ldots \;\leq \;{\left(d_{1}\right)}_{n:n}$ are the ordered distances and compute *μ*_2_ and $\Sigma_{2}$ based on $H_{2}$. Then, $\left|\Sigma_{2}\left|\;\leq \;\right|\Sigma_{2}\right|$ with equality if and only if *μ*_2_ = *μ*_1_ and $\Sigma_{2}=\Sigma_{1}$. If $|\Sigma_{1}|>\;0$, the C-step easily yields a new h-subset with lower covariance determinant than $\Sigma_{1}$, i.e., more “concentrated” than $\Sigma_{1}$. C-steps can be iterated until a stopping criterion is satisfied. The stopping criterion is when $\left|\Sigma_{new}\right|=0$ or when $\left|\Sigma_{new}\left|=\right|\Sigma_{old}\right|$. For the first case, since the covariance matrix is singular, the new candidate subset is rejected and the previous subset is restored as an optimal solution. The sequence of determinants obtained in this way must converge in a finite number of steps because one has only a finite number of h-subsets. However, the final calculation of $\left|\Sigma_{new}\right|$ could not converge to the global minimum of the MCD objective function. Therefore, an approximation of the MCD solution is obtained by randomly selecting several subsets $H_{1}$ as initial guesses, applying the C-steps to each and, in the end, keeping the one with the lowest determinant.

### 2.2. Threshold Estimation Based on Extreme Value Statistics

Dealing with unsupervised methods, where data from a damaged state are not used to help with the damage detection process, the selection of an appropriate threshold to discriminate between normal and not normal behaviors of a structure is difficult to determine. Many research studies make use of statistical process control (SPC) for unsupervised damage detection to monitor such parameters as sample mean and standard deviation [47,48]. SPC is based on the assumption that the distribution of multivariate data is Gaussian, with the MSD subsequently approximated by a chi-squared distribution with p degrees of freedom, where p is the dimension of the feature vector. Therefore, the selection of control limits is the same as the quantile of the underlying distribution associated with the desired confidence level. However, this assumption of normality may result in an improper description in the behaviour of the extreme values of the data, which are associated with the tails of the distribution. As the problem of damage identification specifically focuses attention on these tails, an approach based on Monte Carlo simulation and extreme value statistics [13] is adopted in this paper for a proper threshold setting. The procedure to calculate the threshold is described in the following:

A p×n matrix is created (p is the number of dimensions and n the number of observations), where each p-dimensional observation is generated from a normal distribution, having zero mean and unit standard deviation.The desired discordancy measure (i.e., MSD or MCD) is calculated for all the observations, where mean and covariance are estimated depending on the selected classical or robust methods. The largest value for each matrix is stored.The process is repeated for a large number of iterations in order to create a vector of extreme distances. Then, all the values are sorted in decreasing order. The threshold value depends on the choice of the critical values α. In the following analysis, α is set equal to 5 per cent, giving a 95 per cent confidence limit.

## 3. Cointegration Basics

### 3.1. Order of Integration and Unit Root Tests

The theoretical foundations behind the concept of cointegration are based on a clear understanding of the ideas of stationarity and non-stationarity. A stationary time series is one whose statistical moments are not dependent on time. In case this assumption does not apply, the process is said to be nonstationary. To quantify the degree of non-stationarity of the time series, the order of integration is introduced. More specifically, if a nonstationary time series $y_{t}$ becomes stationary after differencing it d times, then $y_{t}$ is said to be integrated of order d, denoted as $y_{t}\;\sim \;I\left(d\right)$. Just to give an example, $I\left(1\right)$ defines a series which becomes stationary after differencing it once, while $I\left(0\right)$ indicates a stationary series. Different statistical methods have been proposed to determine the order of integration of a time series. One of the most popular is the Augmented Dickey–Fuller (ADF) test [49], which involves fitting the data to the following regression model:(3)$$∆y_{t}=c+\beta t+\rho y_{t-1}+\overset{p}{\underset{j=1}{\sum }}b_{j}∆y_{t-j}+\varepsilon_{t}$$
where $∆y_{t}=y_{t}\;–\;y_{t-1}$, p is the number of lagged difference terms to correct for higher-order correlations, $\varepsilon_{t}$ is the model residual, c and $\beta_{t}$ represent a constant and a linear trend terms which can be eventually added depending on the complexity of the model. The ADF procedure provides an estimate of the parameters of Equation (3) using the least-squares methods and then tests the null hypothesis of ρ=0. The test statistic on ρ is calculated as:(4)$$t_{\rho }=\frac{\hat{\rho }}{s\left(\rho \right)}$$
where $\hat{\rho }$ is the estimate of parameter ρ and $s\left(\rho \right)$ is the standard error of the estimated parameter. Because the t-statistic does not follow one of the typical distributions (Gaussian, *t*-Student, etc.), it should be compared with the critical values of its asymptotic distribution, whose values are tabulated and provided by Dickey and Fuller, depending on the desired significance level α. If $\left|t_{\rho }\right|\;<\;|t_{cr,\alpha }|$, the null hypothesis is accepted, meaning that $y_{t}$ has a unit root and is $I\left(1\right)$. If the null is rejected, the test is repeated for $∆y_{t}$ and if the hypothesis is then accepted, $y_{t}$ is an $I\left(2\right)$ series. If rejected, the procedure is repeated until the integrated order of the process is established.

In this work, the ADF test is used at first to assess the degree of non-stationarity of the input variables. Then, it is adopted after cointegration to verify that the residual obtained from the regression model becomes stationary. This means that cointegration successfully lowered down the order of integration of the original variables, i.e., removes the common trends which caused signals to be nonstationary.

### 3.2. Linear Cointegration

As previously stated, two or more nonstationary series are cointegrated if a linear combination of them can be found to be stationary. Let $Y_{t}=\;\left(y_{1t},\;y_{2t},\ldots ,y_{nt}\right)$ be a multivariate nonstationary, $I\left(1\right)$ time series. The series are cointegrated if there exists a vector $\beta =\;\left(\beta_{1},\;\beta_{2},\ldots ,\beta_{n}\right)$ such that:(5)$$u_{t}=\beta^{T}Y_{t}=\beta_{1}y_{1t}+\beta_{2}y_{2t}+\ldots +\beta_{n}y_{nt}$$ is a stationary univariate time series. Here, the vector *β* is referred to as cointegrating vector and the linear combination $\beta^{T}Y_{t}$ describes the long-run equilibrium relationship between the time series. Usually for a multivariate series such as $Y_{t}$ there could be more than one possible cointegrating relationship and a number of general approaches are introduced to estimate the cointegrating vectors. Among them, the Johansen procedure represents an efficient maximum likelihood estimator [50], which seeks a white Gaussian residual from the cointegrating relation. This method offers the possibility to estimate multiple cointegrating vectors at the same time and to produce a test statistic to determine the number of cointegrating vectors. An alternative approach to estimate the cointegrating relations is the Engle–Granger (EG) two-step procedure [51]. The first step of this method provides to consider a set of measured variables which have a regression model equation of the form:(6)$$y_{1t}=b_{2}y_{2t}+\cdots +b_{n}y_{nt}+\varepsilon_{t}$$
where the time series have the same order of integration and the cointegrating vector $b=\;\left(1,\;-b_{2},\ldots ,-b_{n}\right)$ is unknown and has to be estimated using ordinary least-squares methods. The regressed variable $y_{1t}$ can be selected arbitrarily from the data set, without loss of generality. The estimated regression parameters are then used to form an error correction model (ECM) [52], which describes the long-run relationships between variables. It is not necessary here to discuss about the nature of the ECM, since a simplified implementation of the EG procedure is sufficient for SHM purposes.

The second step then verifies if the obtained cointegration residual $\varepsilon_{t}$ is stationary using the ADF test. If $\varepsilon_{t}$ is stationary, there is cointegration within the series, i.e., the regression equation has captured all the long-term relationships between variables. In this case, a control chart can be directly used to monitor the behaviour of the residual. On the contrary, if $\varepsilon_{t}$ is nonstationary, there is no cointegration within the series.

In this work, the Engle–Granger procedure is preferred to the Johansen approach to test for cointegration because it offers the advantage of having a highly consistent and more intuitive estimator of coefficients, which have already been demonstrated to fit well in case of SHM applications [33,52].

### 3.3. Main Steps to Run Cointegration within SHM Applications

This section resumes the main steps adopted in this paper to estimate the cointegrating relationships over a given set of monitored variables. The Engle–Granger two-step procedure described above is implemented as a core technique to test for cointegration, under the assumption that linear dependencies exist between variables. The overall method can be extended to any kind of SHM applications having similar characteristics.

Select a set of suitable monitored variables belonging to the same process and sharing common trends.Run the ADF test on the variables to determine the order of integration (this should be the same for all the variables).Split the original data set in two parts, one for training and one for testing. Training data are used to estimate the regression model, while test data are used to check for variations in system behaviour. To obtain a reliable model, training data should not include any damaged conditions, but should include a comprehensive time span and resolution of healthy data under different environmental and operational conditions.Run the ADF test on the model residual to assess its stationarity. If this situation is verified, the linear cointegrating relationship is successfully established and common trends are removed. Then, the cointegration residual represents a good indicator of the health status of the structure and can be used for damage detection purposes.

## 4. Proposed Hybrid Approach for Outlier Discrimination

### 4.1. Identify Leverage Points in Regression

In linear regression, observations are classified as $\left(x_{i},y_{i}\right)$, where $x_{i}$ is the p-dimensional vector of predictors and $y_{i}$ is the one-dimensional vector of the response. When $x_{i}$ is far away from the majority of the distribution of the predictors, that point is identified as a leverage point. Therefore, the definition of a leverage point only depends on the behaviour of $x_{i}$, without considering the response $y_{i}$. If both $x_{i}$ and $y_{i}$ lie far from the region associated with the bulk of the data, this observation becomes a bad leverage point. This point can be very critical, since it attracts or even tilts the classical least square regression. If $\left(x_{i},y_{i}\right)$ is still far, but follows the linear relation, it is called a good leverage point, since it improves the precision of the regression estimates. Therefore, in the end, to distinguish between good and bad leverage points, it is necessary to consider both $x_{i}$ and $y_{i}$ and to know the linear relationship established by the majority of the data.

To facilitate the comprehension of the terminology just introduced, Figure 1 shows the different categories of points that may be found in case of simple linear regression. The majority of the data are regular observations and are indicated by (a). Data belonging to (b) and (d) deviate from the linear pattern and thus are called regression outliers, but points in (c) are not. Both (c) and (d) indicates clusters of leverage points, since their $x_{i}$ values are outlying. As a result, points in (c) are good leverage points, while points in (d) are bad leverage points. Finally, observations in (b) can be categorized as vertical outliers, because they are regression outliers but not leverage points.

### 4.2. Description of the Residual Outlier Map

The main results for the case study analyzed in the next section are presented using a residual outlier map similar to that introduced in [29,53]. In these works, the map displays the least trimmed squares (LTS) residuals [53] versus the robust distances calculated using the MCD estimator on the input variables and it is very useful to classify the observations between regular, good and bad leverage points, according to the adopted regression model. In this paper, residuals are established from cointegration analysis. However, this does not change the general interpretation of the results, as both the methods are based on the same theoretical principle, i.e., linear regression.

In more detail, the computation of robust distances reveals leverage points, but cannot distinguish between good and bad ones, because $y_{i}$ is not considered. At the same time, residuals identify regression outliers $y_{i}$ without telling which of them are leverage points. The residual outlier map combines the previous information to provide a powerful visualization in which the inherent nature of the data is revealed.

An illustrative example of the residual outlier map is reported in Figure 2. The two horizontal red lines represent the threshold for the regression residual and depend on its statistical distribution, while the vertical red line is the threshold for the MCD distances. These threshold values divide the plot into six regions categorized with different labels, according to the behaviour assumed by the observations described above.

## 5. Application to SHM: The Railway Bridge KW51

The case study presented in this study is the KW51 bridge, a steel railway bridge that crosses the Leuven–Mechenel canal close to the city of Leuven, Belgium (Figure 3a). Experimental data used in this work have been acquired by the research group from the Structural Mechanics Section of the KU Leuven University and are available online as a benchmark case study.

The KW51 is a steel single-span arch bridge of bowstring type, with a length of 117 m and a width of 12 m. The two-track deck is suspended from the arch with thirty-two inclined braces and it is made of two main girders stiffened by thirty-three transverse beams. The bridge is supported by four neoprene bearings at its extremities, which directly sit on two concrete abutments.

The KW51 has been monitored since 2 October 2018. In the period from 15 May to 27 September 2019, the bridge was retrofitted to resolve a construction error that emerged during inspections. At the beginning of the retrofitting, the scaffolding was installed on the bridge, adding mass to the structure and, therefore, modifying its modal properties. Then, the retrofitting intervention took place, which consisted of strengthening the bolted connections of the braces with the deck and the arch by means of welded steel plates (Figure 3b).

The monitoring system of the KW51 bridge comprises a dense network of sensors which provides measurements of acceleration on the bridge deck and the arches, strain on the bridge deck and on the diagonals connecting the bridge deck with the arches, strain on the rails, displacement at the bearings, and a set of relevant environmental parameters such as temperature, relative humidity and wind speed.

Acceleration measurements represent the quantities of interest for this study. The acceleration of the bridge deck and the arches are acquired by means of 12 uniaxial accelerometers with a sensitivity of 1000 mV/g (model PCB393B04). The layout of the measurement network of the bridge is shown in Figure 4. Six accelerometers are installed on the bridge deck, where four sensors measure the acceleration in the vertical direction (z) and two sensors measure the acceleration in the lateral direction (y). The remaining six accelerometers are installed at the intersection between the arches and the diagonals, where they measure the accelerations in the lateral direction (y).

Acceleration data are acquired and collected every five minutes using a National Instrument (NI) data acquisition system, made of a NI DAQ 9178 chassis and three NI 9234 ICP modules, which account for signal conditioning and A/D conversion. A sampling frequency of 1651.6 Hz is selected.

Ambient vibration data have been used to perform an automated operational modal analysis (OMA) on an hourly basis. The result is the availability of the evolution of the modal parameters (natural frequencies, damping ratios and mode shapes) of the railway bridge over time. For a complete description about the measurement setup, data processing and the available data characteristics readers can refer to [36]. For the purposes of this paper, the four natural frequencies related to the vertical global modes of the bridge deck are considered, referred to as $f_{1},\;f_{2},\;f_{3}$ and $f_{4}$ in Figure 5a. Data samples relate to a period of 15 months, from 2 October 2018 to 15 January 2020. These set of frequencies are of particular interest since they are the only identifiable by OMA with a good success rate also during the retrofitting process, which starts at observation 2675. Missing data that are present in the original dataset are removed as a data pre-processing procedure.

The observations considered for the analysis are classified into five scenarios associated with five distinct structural conditions. To better understand the behaviour assumed by the natural frequencies during these periods, the upper panel of Figure 5b offers an in-depth view of the evolution over time of the second natural frequency $f_{2}$, where the different scenarios are highlighted with different colors. Air temperature readings from the same period are plotted in the lower panel.

Scenarios S1 and S2 refer to the normal conditions of the structure, where the bridge is exclusively influenced by environmental and operational parameters. It can be noted here that the variability of data points is mainly driven by the influence of temperature. In particular, two large peaks are clearly visible in S2 between samples 1404 and 1530. These fluctuations are also observed in the behaviour of other bridges [34,55] and they are highly related to very cold periods, where the temperature decrease below 0 °C causes a stiffness increase due to the freezing of the asphalt layer of the bridge deck.

The retrofitting period is divided into three main regions. The first region (S3) is where a slight decrease in the natural frequency value owing to the presence of the first part of the scaffolding (i.e., added mass) on the bridge is observed; however, its effect on the modification of the structural properties is negligible and EOVs still remain the major source of variation in natural frequencies. The second region (S4) is related instead to the evolution of the retrofitting activities, where all of the bolted connections between the diagonal braces and the deck and arch were strengthened, resulting in a gradual stiffness increase and therefore in a gradual increase in the natural frequencies. Finally, the third region (S5) belongs to the period after completing the retrofitting, where a permanent shift of the natural frequency values is observed with respect to the initial conditions. It should be stressed that, even when structural modifications due to retrofitting occur, natural frequencies still remains sensitive to the influence of temperature, as can be seen from the fluctuations that are present both during and after the retrofitting.

All the scenarios described above are resumed in Table 1, with the corresponding name and data points. Data before retrofitting (S1 and S2) are used for the training of the proposed algorithms in order to fully characterize the structure behaviour during normal operating conditions, while the remaining datasets are used for testing and are assumed to be a proof for structural modifications of different extent.

## 6. Results and Discussions

The univariate MSD and MCD distances are calculated for each of the four natural frequencies in order to highlight the presence of outliers in the data, as shown in Figure 6. For the computation of MCD estimates, a breakdown value h=0.75n has been selected, where n represents the number of observations in the training period. MSD and MCD threshold limits are calculated as described in Section 2.2, given a 95% confidence limit. First of all, one can observe that the Mahalanobis-squared distance is less sensitive to identify outliers. For the specific case of frequency $f_{4}$, the damaged condition has not even been identified. This is because observations related to the freezing period are (correctly) included in the computation of the metrics, since they belong to the normal conditions of the structure, but they biased the estimation of mean and covariance.

On the contrary, the minimum covariance determinant estimator, thanks to its major robustness to outliers, better reveals the presence of abnormal behaviors, resulting in a higher MCD index if compared to the MSD index. The large distances observed in the training period associated with the frequency fluctuations are due to very cold temperatures and represent suitable candidates to test the ability of the proposed combined approach in discriminating outliers due to environmental conditions out of those due to retrofitting. This aspect is investigated in detail in the next section.

Once outliers are revealed, further processing is required in order to transform the original variables into a new feature which is made insensitive to the external influences due to environmental and operational factors and, therefore, create a suitable normal condition training set. This is achieved in this work by establishing a regression model of natural frequencies, based on the concept of cointegration. The basic idea from the point of view of cointegration is that, once cointegration relationship between variables is built, the model error describes their long-term equilibrium. If cointegration relationship is estimated from data when the structure operates under normal conditions, it is expected that natural frequencies are influenced by the same set of external influences and, therefore, their combination produces a model error which is stationary.

The stationarity of residual in this case has a dual function. On one hand it permits us to remove common trends from the original data; on the other, it allows us to check for system variations induced by damage, which may manifest as a deviation from equilibrium conditions. Although the relationship between temperature and each of the natural frequencies of the bridge are found to be nonlinear in case of very low temperatures, relations between natural frequencies themselves within the same period are found to be linear. This is because the different dynamic responses of the structure are driven in a similar manner to the external influence of temperature.

For this reason, linear cointegration is applied in the following analysis. To satisfy cointegration assumptions, all variables included in the regression model must be $I\left(1\right)$. The order of integration for the time series of the four natural frequencies is determined using the ADF test. Results are resumed in Table 2 and show that all the series are nonstationary and integrated of order one at the 95% confidence level.

Then, the two-step Engle–Granger procedure is implemented. Because it is necessary to select one of the natural frequencies at a time to serve as regression response, there exist four possible models, namely, $f_{1}=h\left(f_{2},\;f_{3},\;f_{4}\right)+\varepsilon_{1},\;f_{2}=h\left(f_{1},\;f_{3},\;f_{4}\right)+\varepsilon_{2},\;f_{3}=h\left(f_{1},\;f_{2},\;f_{4}\right)+\varepsilon_{3},\;f_{4}=h\left(f_{1},\;f_{2},\;f_{3}\right)+\varepsilon_{4}\;$, where h represents a function which describes the independent variable as a linear combination of the regressors and $\varepsilon_{i}$ is the model error. Once regression coefficients are estimated, the ADF test is applied to the training residual and the obtained results are reported in Table 3. Since all the ADF test statistics are significantly smaller than the 5% critical value, it is possible to conclude that all residual series are stationary and, therefore, cointegration relationships exist between natural frequencies.

The prediction ability of all the four models are quite similar and tend to have a good fit to the experimental data. However, among all possible models, the one with a residual with the most negative ADF *t*-statistic, namely $f_{2}=h\left(f_{1},\;f_{3},\;f_{4}\right)$, is selected as the candidate for the following analysis. Indeed, this model is the one able to create the most stationary combination of the original variables, i.e., it is the most effective in suppressing the original common trends.

Figure 7 shows the control chart of the residual time series obtained from cointegration. The confidence interval is set to 95%, which corresponds to the range *μ* ± 2*σ*, where *μ* and *σ* are the mean and the standard deviation of the residual calculated from the training data. As one can clearly see, the residual series is very sensitive to structural changes.

During the training period, when the structure is assumed to be in a normal condition, fitted data follow the measured data very well. This is reflected by the residual behavior, which remains stationary, any effect from temperature is effectively deleted and the large fluctuations of the frequency values due to low temperatures are well predicted too. Moving to the testing region, deviations between the model predictions and the experimental data occur. First, an average decrease in residual values between observations 2700 and 2800 is observed, probably related to the added masses of the scaffolding on the bridge, which causes a decrease in the natural frequency with respect to fitted data. Second, from sample 3059 onwards, retrofitting takes place, producing a gradual stiffening of the bridge connections. Natural frequencies increase as well, but with a different relationship with respect to those built during cointegration. As a result, the residual series becomes nonstationary with a sudden jump in magnitude, which strongly points to the occurrence of damage.

At this point, the residual outlier map can be used to classify the observations in the six different regions introduced in Section 4. The plot of the cointegration residual versus MCD robust distances are presented in Figure 8.

Specifically, most of data related to scenarios S1 and S2 are classified as regular (region 3 of the map), as they are characterized by small residual and small MCD distances. This is in agreement to the actual behaviour of the structure, i.e., healthy conditions. Similarly, observations from scenario S3 are associated with region 3, as the presence of the first part of the scaffolding causes only a slight variation in the modal properties of the bridge and thus an increase in the MCD input distances which is negligible, except for few points that appear as vertical outliers. In the latter case, this means that the cointegration relationship is no longer respected. Typically, vertical outliers indicate a model misspecification and thus involve a better comprehension of the main phenomena and a model reformulation. However, this is not the case, since the situation that originates vertical outliers is well-defined and related to an external known cause (temporarily added masses) which cannot be estimated and thus its influence on the modal properties cannot be assessed.

In the map, some points from scenario S2 (very cold temperatures) are attributed to a region of benign variations (region 2). Indeed, they represent good leverage points, as the MCD distances of regressors assume high values, but at the same time, the residual is within the control limits, indicating that the cointegration relationship established during normal conditions is still respected.

On the other hand, observations from scenarios S4 and S5 are mostly related to region 4, i.e., they are bad leverage points. This time, as well as input variables are characterized by high MCD distances, residual amplitude also increases since the equilibrium described by cointegration is violated due to the different behaviour induced by retrofitting. Once again, the residual outlier map is able to separate the observations based on the current state of the structure. Consistent with the development of the retrofitting activities, it is possible to observe that data points related to scenario S4 migrate from region 3 (healthy conditions/negligible changes) to region 4 (damage state/relevant structural modification), while data points after the completion of the retrofitting lie permanently within region 4.

## 7. Conclusions

In this paper, a novel approach based on the use of cointegration and robust outlier detection is described as a means of characterizing outliers in the data. When structures are operating in situ and outside a controlled environment, the measured features may change over time due to the influence of benign reasons, mainly environmental and operational factors that are unrelated to damage. Different outlier detection methods have been used to reveal the presence of anomalies with respect to the baseline conditions, and outliers are often interpreted as critical points to be discarded although they may contain relevant information. On the other hand, when groups of outliers are included in the analysis, they could bias model estimates and the subsequent performance. Consequently, it is of key importance to reveal the nature of outliers according to the underlying processes that generate them. Thanks to this analysis, one can detect benign variability in advance, creating normal conditions clear of external influences, thus enhancing the possibility to identify real damages.

This is exactly the central target of this work, which focuses on explaining that outliers due to environmental and operational variations appears in the data in a different way with respect to those due to damages or structural modifications. The proposed method has been applied on the experimental data of the KW51 railway bridge, which represents a realistic case study subjected to the contemporary influence of EOVs and a gradual structural modification due to retrofitting.

Natural frequencies associated with the vertical modes of the bridge represent the input features for the procedure. Observations are divided in five scenarios associated with different structural conditions. First, data samples from the period before retrofitting, where the structure is assumed to be healthy, are used as training data to establish a cointegration relationship where one frequency (predictor) is expressed as a linear combination of the others (regressors). Data samples acquired during and after the retrofitting are used instead for testing purposes. The stationarity of the training residual obtained after cointegration is assessed through a unit root test and confirms that EOVs trends are effectively removed; at the same time, deviations from stationary conditions are observed in the testing residual after retrofitting which strongly points to the occurrence of damage.

Then, the information from the cointegration residual is used in conjunction with the robust discordancy measures (MCD) of the regressors to create a residual outlier map, which allows us to distinguish the observations in six different regions based on the different structural conditions. Samples related to the healthy state of the structure are correctly classified as regular. Samples measured in correspondence to very cold temperatures are treated as good leverage points and, therefore, associated with a region of “benign” variations; indeed, in this case, even though the MCD distances of the regressors are far from the bulk of the distribution, the cointegration relationship established during normal conditions is still preserved. Finally, it is observed that retrofitting causes a gradual change of the modal properties of the structure which is different from that induced by EOVs. As a result, not only high MCD distances of the regressors are obtained, but also high residual amplitudes, which testify that the equilibrium conditions established by the cointegration relationship are violated. Again, this aspect has been properly captured by the residual outlier map, which identifies these observations as bad leverage points and locates them in a region of “malign” variations.

Future studies should be oriented towards the extension to nonlinear robust regression methods, so as to also include those features characterized by nonlinear relationships.

## Figures and Tables

**Figure 1 sensors-22-02177-f001:**
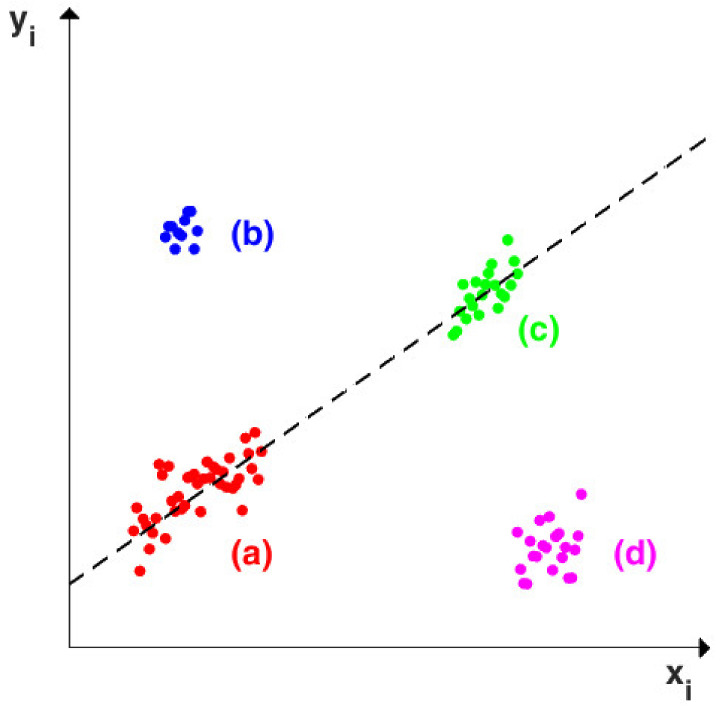
Example of simple linear regression showing (a) regular observations, (b) vertical outliers, (c) good leverage points, (d) bad leverage points.

**Figure 2 sensors-22-02177-f002:**
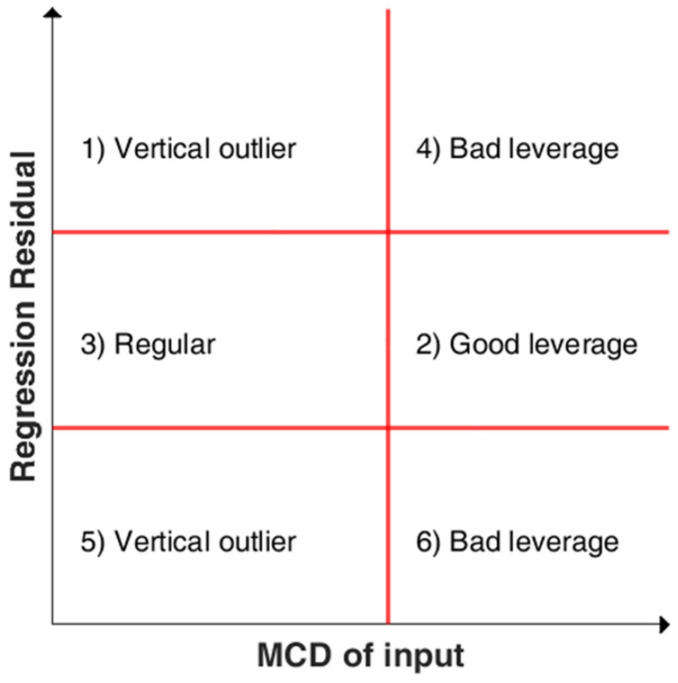
Graphical representation of residual outlier map.

**Figure 3 sensors-22-02177-f003:**
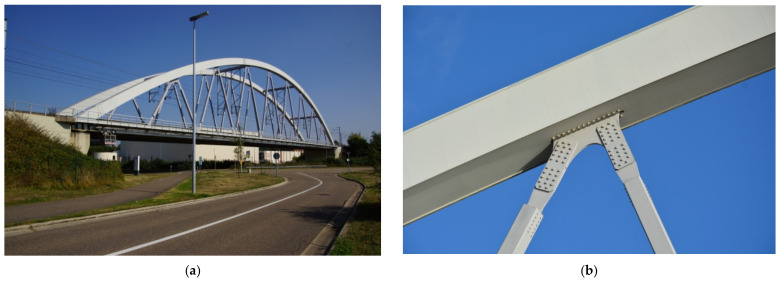
(**a**) The KW51 bridge in Leuven and (**b**) Example of bolted connection between a brace and the arch of the bridge (Reprinted from ref. [54]).

**Figure 4 sensors-22-02177-f004:**
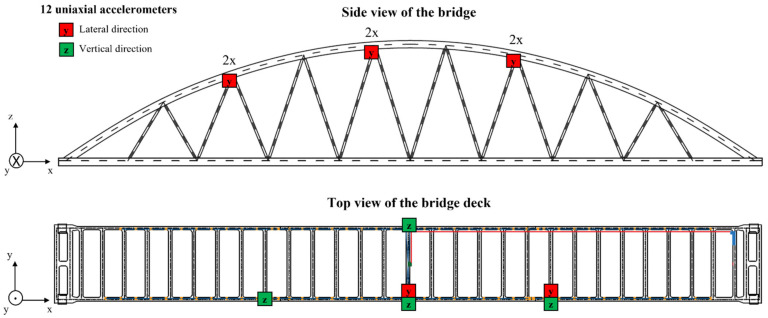
Layout of the acceleration measurement setup of the KW51 bridge. The side view shows the sensors installed on the connections between the diagonals and the arches, while the top view illustrates the sensors installed on the bridge deck. (Adapted from ref. [54]).

**Figure 5 sensors-22-02177-f005:**
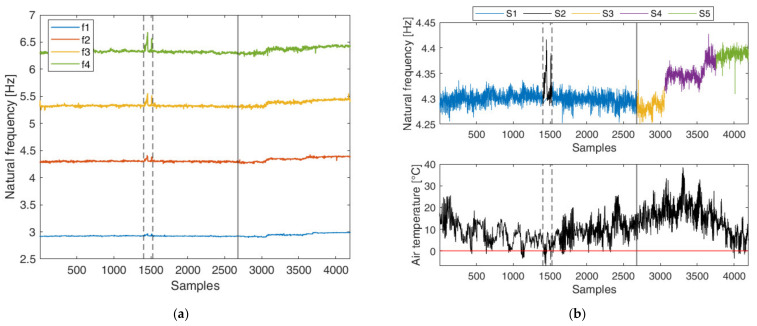
(**a**) Time history (hourly based) of natural frequencies associated with the global vertical modes of the bridge deck. Vertical dashed lines indicate the frost period, while vertical solid line indicates the beginning of retrofitting interventions; (**b**) Upper panel: time history of the second natural frequency of the bridge deck, where the five analyzed scenarios are plotted in different colors; Lower panel: time history of air temperature (horizontal red line indicates air temperature equal to 0 °C).

**Figure 6 sensors-22-02177-f006:**
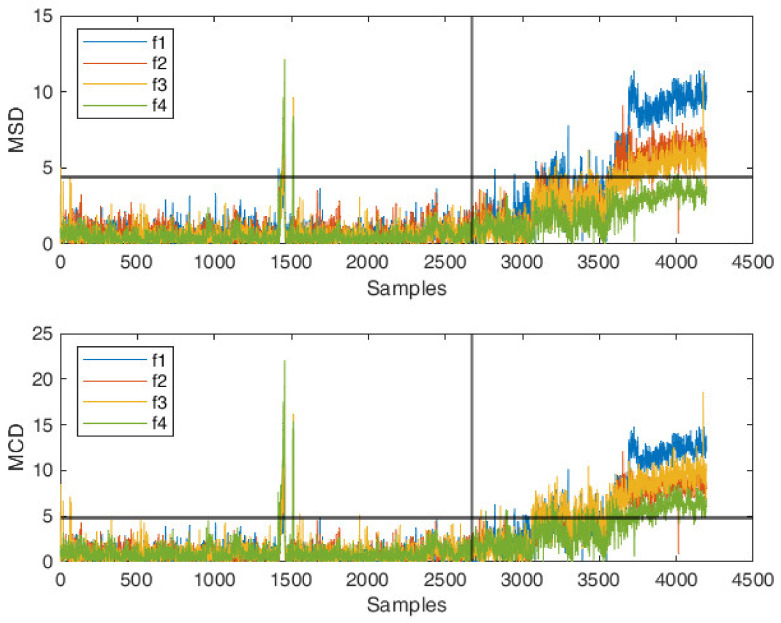
Comparison between univariate MSD and MCD distances for the four natural frequencies.

**Figure 7 sensors-22-02177-f007:**
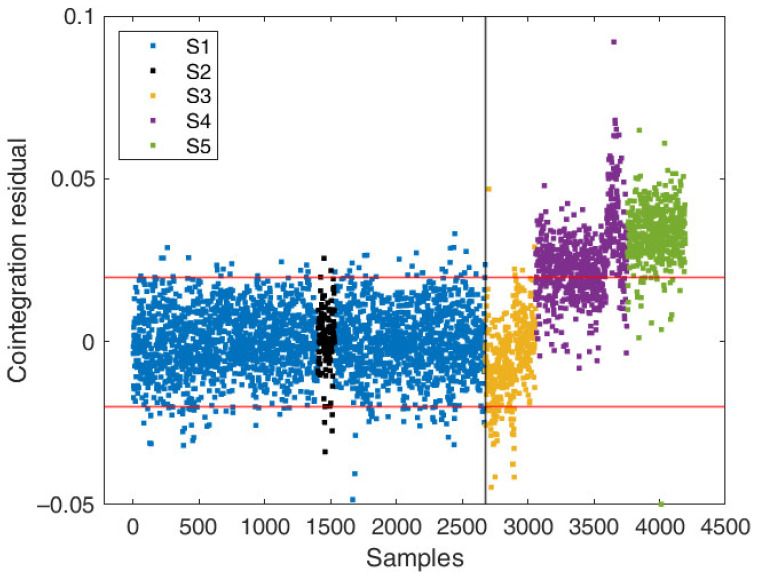
Control chart of cointegration residual from model $f_{2}=h\left(f_{1},\;f_{3},\;f_{4}\right)$. The upper and lower red lines represent the training residual mean plus or minus two times the standard deviation.

**Figure 8 sensors-22-02177-f008:**
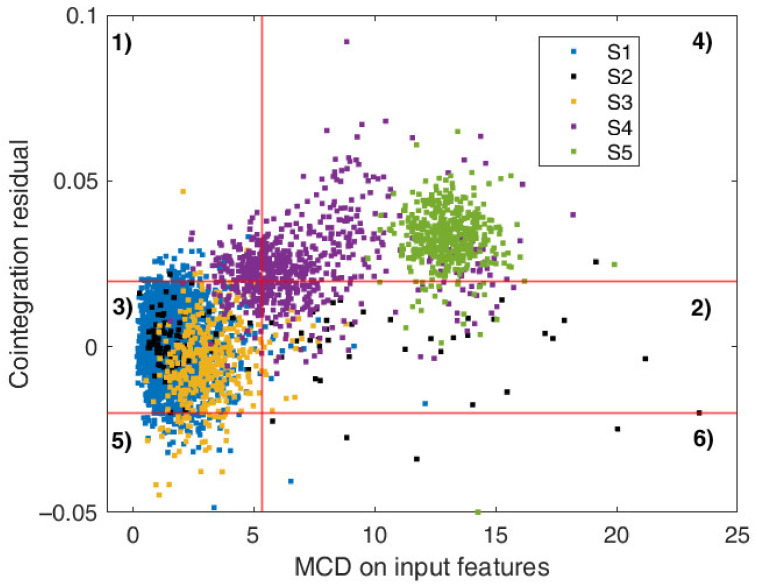
Plot of cointegration residual versus MCD robust distance for model $f_{2}=h\left(f_{1},\;f_{3},\;f_{4}\right)$. Numbers from 1 to 6 refer to the different regions of the residual outlier map described in Section 4.2.

**Table 1 sensors-22-02177-t001:** Description of the different analyzed scenarios.

Scenario	Description	Data Points
S1	Before retrofitting	1–1403; 1531–2674
S2	Before retrofitting (cold temperatures)	1404–1530
S3	Retrofitting: 1st stage	2675–3058
S4	Retrofitting: 2nd stage	3059–3755
S5	Post retrofitting	3756–4196

**Table 2 sensors-22-02177-t002:** ADF test results for the four natural frequencies series.

Variables	ADF *t*-Statistic	5% Critical Value	Stationarity?
$f_{1}$	0.041	−1.942	NO
$f_{2}$	0.030	−1.942	NO
$f_{3}$	−0.091	−1.942	NO
$f_{4}$	−0.072	−1.942	NO

**Table 3 sensors-22-02177-t003:** ADF test results for the training residuals.

Variables	ADF *t*-Statistic	5% Critical Value	Stationarity?
$\varepsilon_{1}$	−14.491	−1.942	Y
$\varepsilon_{2}$	−18.629	−1.942	Y
$\varepsilon_{3}$	−17.557	−1.942	Y
$\varepsilon_{4}$	−12.301	−1.942	Y

## Data Availability

Not applicable.

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
