# Peer review of "Combined Use of Cointegration Analysis and Robust Outlier Statistics to Improve Damage Detection in Real-World Structures"

_sensors, 2022, doi:10.3390/s22062177_

Round 1

Reviewer 1 Report

This new and interesting method is very necessary for evaluation the behavior in situ of different types of structures. Due to the environmental interferences, some behavior are not related to damage. So , this method is more appropriate and it can be extended to other types of structures.

Author Response

Dear Reviewer 1,

thank you very much for the time and the effort that you spent to provide your valuable feedback. We revised the entire manuscript and we improved both English language and style at the best of our knowledge.

Reviewer 2 Report

The article presents a methodology that aims to reduce the influence of outliers for the detection of “damage” in structures (structural novelties). This is an interesting study, but it deserves some clarification:

- Considering that the vast majority of current articles that address the theme of the manuscript address detection/location/quantification of damage, the reviewer understands that arguments are necessary to justify and highlight the importance of publishing articles that only detect damage, especially in actual structures.

A crucial point that was unclear to the reviewer is how the authors interpret the five analyzed scenarios. Are there five different scenarios? (5 distinct structural conditions and, therefore, the presented algorithm must allow differentiating these five conditions) ? Alternatively, are the small structural changes caused by retrofitting first/second stages considered benign (similar to effects caused by ambient loading), and therefore should they not be identified as distinct stages of “damage”?
- If the authors consider that there are 5 “damage” scenarios (or structural states) to be identified, it is observed that only scenario five was clearly distinguished from the others. Thus, it is noted that the proposed methodology did not perform satisfactorily in most cases, and this should be better detailed and justified in the text (figures 5 and 6);
Otherwise i.e., some scenarios are of slight changes (analogous to changes caused by environmental effects). In this case, should these scenarios be identified as belonging to the same structural behavior? Please, clarify it.

- Natural frequencies peaks at temperatures < 0 did not appear for samples > 4000 (Figure 3). Why?

- Minor comments:
- the “4Error! Reference source not found.” must all be removed
- Fonts must be standardized. Example page 5, lines 204 to 208….H1 (H in italics and 1 normal);…H1(H in italics and 1 as a subscript); H2 (H without italics and 2 normal).

Reviewer 3 Report

Thank you for the opportunity to review this manuscript. However, there are some inconsistencies between the discussions provided by the authors. The authors mentioned data fusion. However, it seems the authors used the technique of data representation.

The authors have used the technique which came from reference number 25. However, it was an old paper. Recently the researcher used different machine learning algorithms that can overcome the mentioned problem.

Therefore, the authors should validate the methodology. 

Even, new NDT techniques that are used in the SHM system can detect damage with the environmental effects. The authors have technically ignored this point.

The language of the manuscript needs some improvement.

Reviewer 4 Report

Paper title:

Combined use of cointegration analysis and robust outlier statistics to improve damage detection in real-world structures

Authors

S. Turrisi, E. Zappa, A. Cigada

 The submitted work presents an interesting approach which allows to visualize and classify outliers in structural health monitoring data.

In the submitted study the KWS1 bridge is used an application paradigm regarding the applicability and the effectiveness of the proposed methodology. Nevertheless, more details about the configuration and the geometrical characteristics have to be included in the manuscript along with a photograph of the bridge.

Moreover, details about the network of the sensors would be really helpful for the reader. In line 394 it is stated that for the available data characteristics the reader has to take a look at [3]. However, in order to have a stand alone paper the authors have to include in the submitted manuscript the main characteristics of the measurement setup and the necessary data characteristics.

In figure 3a. What the meaning of f1, f2, f3, f4? Probably the names of sensors at specific positions of the bridge. Therefore, it is apparent that an outline of the instrumentation setup is absolutely necessary in the revised manuscript.

The conclusions include interesting qualitative comments. Thereupon, quantitative results and comparisons may improve this final paragraph.

A notation list is definitely required (abbreviations included).

Final Conclusion

The submitted paper is a useful work that may be accepted after a careful revision.

Round 2

Reviewer 2 Report

The authors have sufficiently improved the quality of the article. I recommend it for publication.

Reviewer 3 Report

The authors have answered all the queries, therefore the manuscript can be publishable.

Reviewer 4 Report

Paper title:

Combined use of cointegration analysis and robust outlier statistics to improve damage detection in real-world structures

Authors: Turrisi, E. Zappa, A. Cigada

In the revised manuscript and at the beginning of section 5 (lines 407-423), the authors have added a sufficient description of the main geometrical characteristics and design properties of the bridge along with two photographs, (Figure 3 of the revised manuscript).

Furthermore, some information about the network of the sensors has also been added in the manuscript in section 5 (lines 426-442).

The conclusions include quantitative results and comparisons.

An informative notation list has been added, too.

Final Conclusion

In general, the authors have successfully responded to the reviewer’s comments.

The submitted paper is a useful work and can be accepted as is.